# Frequency-Agility-Based Neural Network with Variable-Length Processing for Deceptive Jamming Discrimination

**DOI:** 10.3390/s25175471

**Published:** 2025-09-03

**Authors:** Wei Gong, Renting Liu, Yusheng Fu, Deyu Li, Jian Yan

**Affiliations:** University of Electronic Science and Technology of China, Chengdu 611731, China; 202322010911@std.uestc.edu.cn (W.G.); liurt@uestc.edu.cn (R.L.); yushengf@uestc.edu.cn (Y.F.); ncustudwntldy@163.com (D.L.)

**Keywords:** frequency agility, multistatic radar, deception jamming discrimination, neural network, variable-length processing

## Abstract

With the booming development of the low-altitude economy and the widespread application of Unmanned Aerial Vehicles (UAVs), integrated sensing and communication (ISAC) technology plays an increasingly pivotal role in intelligent communication networks. However, low-altitude platforms supporting ISAC, such as UAV swarms, are highly vulnerable to deception jamming in complex electromagnetic environments. Existing multistatic radar systems face challenges in processing slowly fluctuating targets (like low-altitude UAVs) and adapting to complex electromagnetic environments when fusing multiple pulse echoes. To address this issue, targeting the protection needs of low-altitude targets like UAVs, this paper leverages the characteristic of rapid amplitude fluctuation in frequency-agile radar echoes to analyze the differences between true and false targets in multistatic frequency-agile radar systems, particularly for slowly fluctuating UAV targets, demonstrating the feasibility of discrimination. Building on this, we introduce a neural network approach to deeply extract discriminative features from true and false target echoes and propose a neural network-based variable-length processing method for deception jamming discrimination in multistatic frequency-agile radar. The simulation results show that the proposed method effectively exploits deep-level echo features, significantly improving the discrimination probability between true and false targets, especially for slowly fluctuating UAV targets. Crucially, even when trained on a fixed number of pulses, the model can process input data with varying pulse counts, greatly enhancing its practical deployment capability in dynamic UAV mission scenarios.

## 1. Introduction

The rapid evolution of modern warfare and the low-altitude economy has spurred the extensive deployment of UAV platforms. Concurrently, electronic warfare equipment, particularly active deception jamming generated by Digital Radio Frequency Memory (DRFM), has been widely employed due to its low cost and high effectiveness, posing significant threats in combat and to low-altitude ISAC systems through malicious interference [1,2]. Recent studies have further demonstrated its adaptability to complex low-altitude environments [3]. DRFM techniques can precisely replicate, modulate, and retransmit intercepted radar signals, generating highly deceptive false targets in multiple dimensions such as range [4] and velocity [5]. These false targets severely threaten the collaborative sensing of UAV swarms, the security of communication links, and the detection and tracking capabilities of low-altitude defense systems against genuine UAV targets [6].

Current deception electronic countermeasure strategies can be classified into monostatic radar and multistatic radar systems. For monostatic radar countermeasures, extensive research has emerged in areas such as transmitted signal optimization [7], waveform agility [8], resource scheduling [9], and intelligent anti-jamming techniques [10,11]. However, monostatic radars suffer from limited perspectives and constrained observation resources, resulting in restricted Electronic Counter-Countermeasures (ECCM) capabilities. Especially in low-altitude ISAC systems, deceptive jamming simultaneously disrupts both sensing and communication link functionalities [12]. Consequently, multistatic radar strategies that leverage networked radars to acquire multi-perspective and multi-dimensional resources represent a key developmental trend for enhancing ECCM capabilities [13,14], especially in safeguarding low-altitude airspace (e.g., UAV traffic management and border surveillance) and ensuring the robustness of ISAC networks [15,16]. This trend is further amplified by the evolving landscape of 6G communications [17,18], which emphasizes ubiquitous connectivity, integrated sensing and communication (ISAC), and the integration of terrestrial and non-terrestrial networks, creating a critical need for resilient and adaptive anti-jamming solutions within such complex, heterogeneous environments [19,20].

Within networked radar systems targeting low-altitude ISAC applications, signal-level fusion methods extract and integrate subtle echo signal characteristics from multiple receiving stations, significantly improving information utilization compared to data-level fusion [21,22] with recent advances in multi-dimensional processing [3,23]. Existing techniques primarily exploit differences in spatial scattering characteristics between targets and jamming. Reference [24] achieved rapid-fluctuating target discrimination using multi-echo correlations under spatial independence, but their performance degrades for slow-fluctuating targets (e.g., hovering or low-speed UAVs). Addressing this, [25] proposed a single-Pulse Repetition Interval (PRI) discrimination scheme using spatial instead of temporal information. Furthermore, energy distribution disparity was incorporated for multi-dimensional discrimination [26].

These approaches typically assume impractical 100% target detection probability [27]. Reference [28] introduced a jamming feature matrix to overcome this, but its dependence on precise matrix estimation limits utility in dynamic UAV jamming scenarios. Ref. [29] incorporated polarization scattering without resolving the estimation challenge. To reduce manual feature extraction uncertainties, reference [30] utilized convolutional neural networks (CNNs) to extract detailed features, improving accuracy. Critically, however, the method in [30] requires repeated training for different Pulse Repetition Interval (PRI) counts, rendering it infeasible to adapt to the potentially dynamic pulse number variations inherent in UAV missions, thus severely hindering practical deployment. While recent advances in attention-enhanced networks have shown promise in improving robustness for low-SNR signal recognition [31,32], they remain constrained by fixed-length input requirements.

Low-altitude UAV targets are susceptible to complex clutter interference, with their echoes exhibiting typical slowly fluctuating characteristics [33]. Traditional discrimination algorithms based on spatial independence suffer from significant performance degradation for such targets, while existing neural network approaches struggle to adapt to dynamic pulse variations. To enhance multistatic radar discrimination against slow-fluctuating UAV targets under deception jamming in complex low-altitude environments and improve applicability in dynamic ISAC contexts, this paper first incorporates frequency agility [23]. We investigate target correlation characteristic variations in a multistatic frequency-agile radar system, validating the technology’s effectiveness in improving existing multi-pulse correlation-based algorithms, particularly for slow-fluctuating UAV targets. Addressing the practical challenges of UAV networked anti-jamming, we then propose a neural network-based variable-length processing method for jamming discrimination, integrating segmentation and data padding mechanisms. The core advantage of this approach is that a single training instance with a fixed pulse count enables flexible processing of inputs with varying PRI numbers. This effectively resolves the critical bottleneck of repeated training required by existing neural methods like [24] in the context of variable UAV mission profiles, significantly enhancing the deployment feasibility of neural network-based anti-jamming techniques in real-world low-altitude ISAC systems.

Recent variable-length architectures for radar deception discrimination face several critical limitations [34,35]. Attention models [31,32], while offering sequence-length adaptability, suffer from prohibitively high $O(Q^{2})$ memory complexity, making them impractical for long pulse sequences. RNNs and LSTMs [36] maintain temporal coherence but struggle with vanishing gradients when processing slow-fluctuating targets over extended coherent processing intervals (CPIs). The spatial correlation method mentioned in reference [24] also fails for slowly fluctuating targets. Transformers [37], despite enabling global feature extraction, rely heavily on precise positional encoding, which can be significantly distorted by radar phase noise.

In contrast, our proposed segmentation-and-masking approach addresses these challenges effectively. By processing fixed-length segments, it achieves linear complexity (O(Q)), drastically reducing computational overhead. By integrating frequency-agility-induced RCS fluctuations with a segmentation-based neural architecture, we resolve the core conflict between pulse-length adaptability and slow-fluctuating target discrimination. Additionally, our method enhances phase-noise robustness through amplitude-centric feature learning, as detailed in Section 3.2. This combination of efficiency and resilience makes our solution particularly well-suited for real-world radar deception discrimination tasks.

The structure of the remaining sections is as follows: In Section 2, we establish the signal model for detecting active false targets in a multiple station frequency-agile radar system. In Section 3, we discuss the processing flow of the proposed variable-length neural network-based method. In Section 4, we present the simulation results and analyses to validate the effectiveness and superiority of the proposed approach. In Section 5, we comprehensively examine methodological novelty, practical implications, and robustness guarantees through theoretical and comparative discussion. Finally, Section 6 concludes the paper.

## 2. Signal Model

As illustrated in Figure 1, this chapter considers a heterogeneous multiple station radar system consisting of one inter-pulse frequency-agile transmitter station and *M* inter-pulse frequency-agile receiver stations, The transmitter station is located at $\left(x^{t},y^{t}\right)$, while the receiver stations are positioned at $\left(x_{m}^{r},y_{m}^{r}\right)$, where m=1,2,…,M. Multiple radars will simultaneously focus on the joint detection area. The system contains *K* real targets (including stealth targets). It is assumed that the self-defense jammers carried by these real targets will deploy deceptive jamming against all receiver stations. Each receiver station receives *D* different false target measurements. Consequently, the total number of targets becomes W=K+L, where L=DK.

This article assumes that the time synchronization and spatial synchronization of the multi-base radar system have been completed [38]. At the q-th pulse moment, the actual echo signal received by the *m*-th radar receiving station can be expressed as(1)$$r_{m}\left(t,q\right)=\sum_{k=1}^{K}e_{m,k}\left(t,q\right)+\sum_{k=1}^{K}\sum_{d=1}^{D}j_{m,k,d}\left(t,q\right)+n_{m}\left(t\right)$$

In the formula, q=1,2,…,Q denotes the number of pulses within a coherent processing interval (CPI). $e_{m,k}\left(t,q\right)$ represents the echo signal of the *k*-th true target received by the *m*-th receiving station. $j_{m,k,d}\left(t,q\right)$ denotes the interfering echo signal of the *d*-th false target received by the *m*-th receiving station in the direction of the *k*-th true target. $n_{m}\left(t\right)$ corresponds to the Gaussian-distributed noise signal.

The echo signal $e_{m,k}\left(t,q\right)$ of the true target can be expressed as(2)$$e_{m,k}\left(t,q\right)=\alpha_{m,k}^{q}s\left(t-\tau_{m,k}\right)\exp\left\{-j2\pi f_{q}\left(t-\tau_{m,k}\right)\right\}$$
where $\alpha_{m,k}^{q}=\lambda_{q}\sigma_{m,k}^{q}\sqrt{P_{T}^{q}G_{T}G_{Rm}}/\left(4\pi \sqrt{4\pi }R_{T}R_{m,k}\right)$ denotes the complex amplitude of the echo signal from the *k*-th true target received by the *m*-th receiving station. $\lambda_{q}=c/f_{q}$ denotes the wavelength of the carrier signal for the *q*-th inter-pulse frequency-agile pulse. *c* denotes the speed of light. $f_{q}=f_{0}+\left(q-1\right)\Delta f$ denotes the carrier frequency of the *q*-th inter-pulse frequency-agile signal. $f_{0}$ denotes the initial carrier frequency. Δf denotes the frequency step between adjacent carrier frequencies. $\sigma_{m,k}^{q}$ denotes the radar cross-section (RCS). Due to frequency agility of the signal, it can be assumed that $\sigma_{m,k}^{q}$ follows a complex Gaussian distribution $\sigma_{m,k}^{q}∼\mathrm{CN}\left(0,\zeta_{m,k,q}^{2}\right)$. $P_{T}^{q}=P_{T}\lambda_{Q}^{2}/\lambda_{q}^{2}$ denotes the transmit power of the *q*-th inter-pulse frequency-agile signal at the transmitting station. $P_{T}$ denotes the initial transmit power of the transmitting station. During the transmission phase, power compensation is applied to the amplitudes of different frequency-agile pulses to counteract the amplitude variations induced by frequency agility. $G_{T}$ denotes the transmit antenna gain, while it represents the receive gain of the *m*-th receiving station. $R_{T}$ denotes the distance from the transmitting station to the *k*-th true target. $R_{m,k}$ denotes the distance from the *k*-th true target to the *m*-th receiving station. s(t) denotes the baseband signal. $\tau_{m,k}=R_{m,k}/c$ denotes the echo time delay from transmission to the *m*-th receiving station for the *k*-th true target.

The deceptive jamming signal $j_{m,k,d}\left(t,q\right)$ can be expressed as(3)$$j_{m,k,d}\left(t,q\right)=\beta_{m,k,d}^{q}s\left(t-t_{m,k,d}^{\prime }\right)\exp\left\{-j2\pi f_{q}\left(t-t_{m,k,d}^{\prime }\right)\right\}$$
where $\beta_{m,k,d}^{q}=\gamma_{d}\lambda_{q}\sqrt{P_{J}G_{Rm}}/\left(4\pi R_{m,k}\right)$ denotes the complex amplitude of the *d*-th false target echo signal received by the *m*-th receiving station in the direction of the *k*-th true target. $\gamma_{d}$ represents the random complex amplitude modulation introduced by the jammer. Following Reference [39], $\gamma_{d}$ can be assumed to follow a Gaussian distribution. $P_{J}$ indicates the signal power of the jammer. $t_{m,k,d}^{\prime }=\tau_{m,k}+d\Delta \tau$ corresponds to the actual time delay of the active false target. Δτ represents the time delay interval between active false targets, where the corresponding range interval defines the false target density.

The internal noise of the radar receiver is complex Gaussian white noise, and the noise signals at different receiving stations can be considered mutually independent. Given the noise power as $\sigma_{m}^{2}$, the noise signal $n_{m}\left(t\right)$ can be expressed as(4)$$n_{m}\left(t\right)∼\mathrm{CN}\left(0,\sigma_{m}^{2}\right)$$

## 3. Interference Detection Method Based on Variable-Length Neural Network Processing

### 3.1. Frequency-Agile Radar Echo Signal Analysis

Compared with conventional multiple station radar signals, the frequency-agile characteristics of multiple station frequency-agile radar signals will introduce inter-pulse echo phase variations, with the modulation effect being target-position-dependent. To maximize the correlation between false target signals, phase compensation of echo signals becomes essential. While the existing literature [40,41] has thoroughly investigated phase compensation for frequency-agile signals, this paper primarily focuses on discrimination methods. Therefore, we assume the signals have been properly compensated.

Each receiving station can extract the measurement information of the same target and the collection of slow-time sampling sequences within one CPI after performing complex mixing, low-pass filtering, angle estimation, pulse compression, and constant false alarm rate (CFAR) detection on the received echo signals. After range-phase compensation, the coherent slow-time complex envelope sequence of the *w*-th target received by the *m*-th receiving station within one CPI is given by(5)$$\begin{array}{cc} {\tilde{S}}_{m,w}=S_{m,w}+Z_{m} & \left(w=1,2,\ldots ,W\right) \end{array}$$
where $S_{m,w}$ denotes the noise-free multiple station radar complex envelope sequence. $Z_{m}∼\mathrm{CN}\left(0,\sigma_{m}^{2}I_{Q\times Q}\right)$ represents the noise signal sequence. $I_{Q\times Q}$ is an identity matrix of dimension Q×Q. Given the noise independence across receiving stations, we have(6)$$\begin{array}{cc} E[{\left(Z_{m1}\right)}^{H}Z_{m2}]=0 & \left(m1\neq m2\right) \end{array}$$

The backscattering cross-section of a real target at different times is related to Δf, the frequency step between adjacent carrier frequencies of the transmitting station. To enhance the variation in the target’s RCS across pulses, the frequency step Δf between adjacent transmitted carrier frequencies should be sufficiently large. According to the frequency independence condition of echoes [42], the complex amplitudes $S_{m,w}^{q}$ of the real target’s echo signals at different pulse times within one CPI are mutually independent. This results in rapid fluctuation of the multi-pulse echo complex amplitudes, denoted as $E\left[S_{m,k}^{q1}S_{m,k}^{q2}\right]=0\left(\forall q1\neq q2\right)$. Assuming the real target’s echo power is $\sigma_{m,w}^{2}$, the compensated complex amplitude signal $S_{m,w}^{q}$ can be modeled as following a complex Gaussian distribution:(7)$$S_{m,w}^{q}∼\mathrm{CN}\left(0,\sigma_{m,w}^{2}\right)$$

After the aforementioned processing, the actual slow-time complex envelope sequences ${\tilde{S}}_{m,w}$ from each receiving station are transmitted to the fusion center for joint detection, where a discrimination algorithm is applied to distinguish between true and false targets.

To verify the feasibility of the discrimination method for slow-fluctuating targets using frequency-agility technology, this paper conducts simulation experiments on the actual acquired slow-time complex envelope sequences ${\tilde{S}}_{m,w}$ under two scenarios: without frequency-agility (using the multi-pulse correlation-based jamming discrimination method from reference [39] for non-fluctuating single-frequency true targets) and with frequency-agility technology, with the results shown in Figure 2.

As can be observed, the FAMC method demonstrates superior true target discrimination performance compared to the MC method in Reference [39]. This validates that frequency-agility technology significantly enhances the discrimination capability for non-fluctuating targets.

### 3.2. Neural Network Input Data Preprocessing

The actual target echo signals contain abundant hidden deep-level features that were not fully exploited. The correlation-based discrimination method in Reference [39] only provides a partial solution for distinguishing true target echoes from jamming signals. To better extract the intrinsic differences between genuine and false targets, the application of neural networks offers a new feasible approach with superior feature extraction capabilities.

To simplify pulse quantity control, this study constructs time–space 2D sample matrix data ${\tilde{S}}_{matrix}$ by aggregating slow-time envelope sequences from multiple radar stations observing either the same true target or the same false target, as formulated in Equation (Equation 5).

Let ${\tilde{S}}_{m,w}\in C^{Q\times 1}$ denote the *column vector* of slow-time complex envelopes for target *w* at station *m*:(8)$${\tilde{S}}_{m,w}={\left[{\tilde{S}}_{m,w}^{1},{\tilde{S}}_{m,w}^{2},\ldots ,{\tilde{S}}_{m,w}^{Q}\right]}^{T}$$(9)$${\tilde{S}}_{\mathrm{matrix}}=\left[{\tilde{S}}_{1,w}|{\tilde{S}}_{2,w}|\ldots |{\tilde{S}}_{M,w}\right]={\left[\begin{array}{cccc} {\tilde{S}}_{1,w}^{1} & {\tilde{S}}_{2,w}^{1} & \ldots & {\tilde{S}}_{M,w}^{1}\\ {\tilde{S}}_{1,w}^{2} & {\tilde{S}}_{2,w}^{2} & \ldots & {\tilde{S}}_{M,w}^{2}\\ \vdots & \vdots & \ddots & \vdots \\ {\tilde{S}}_{1,w}^{Q} & {\tilde{S}}_{2,w}^{Q} & \ldots & {\tilde{S}}_{M,w}^{Q} \end{array}\right]}^{T}$$

Since the signal amplitudes at different receiving stations are affected by various factors such as distance and power, we perform station-wise normalization on the original time–space 2D sample matrix to enhance the generalization capability of the neural network model. The normalized sample data is denoted as ${\hat{S}}_{matrix}$:(10)$${\hat{S}}_{matrix}=\left[\begin{array}{c} {\hat{S}}_{1,w}\\ {\hat{S}}_{2,w}\\ {\hat{S}}_{3,w}\\ {\hat{S}}_{4,w} \end{array}\right]$$

In slow-time envelope sequence data, the variable pulse count *Q* in practical scenarios leads to input 2D sample matrices with varying widths. To enhance the model’s discrimination capability for slow-time sequences of different lengths, this section introduces a data padding method to standardize all sequences to a uniform processing length.

The network architecture employing data padding demonstrates inherent capability for variable-length input processing, effectively handling slow-time sequences of diverse lengths. However, two critical limitations emerge when processing exceptionally long sequences: the optimal padding length becomes difficult to determine empirically, and excessive padding dilutes the effective feature concentration in input data, potentially obscuring crucial discriminative patterns. These constraints motivate the introduction of a data segmentation preprocessing mechanism to simultaneously address the padding length determination challenge and enhance the method’s generalization across varying data lengths.

Data segmentation divides continuous sample data with different pulse counts into fixed-length input samples. Based on whether the samples overlap, it is classified into overlapping segmentation and non-overlapping segmentation. As shown in Figure 3, to reduce the number of segmentation operations while effectively improving classification accuracy, this section first performs non-overlapping segmentation and pads the final insufficient segment with data from the end of the previous segment to reach the preset fixed length.

Under the data segmentation preprocessing operation, each model independently predicts the segmented data, generating multiple labels. To integrate the prediction results from segmented data and achieve final discrimination between true and false targets, this section adopts a voting strategy from ensemble learning, where the class with the majority votes among the model predictions is selected as the final result.

Additionally, since the original sample data is complex-valued, the real and imaginary parts of the segmented input data are concatenated row-wise (as shown in Figure 4) to both simplify the neural network processing mechanism and preserve the characteristics of the complex data.

### 3.3. Network Architecture

As shown in Figure 5, the constructed neural network model consists of an input layer, a mask layer, convolutional and max-pooling layers, a dot-product self-attention mechanism with residual connections, a flatten layer, global average pooling and fully connected layers, and an output layer.

The signal processing pipeline is rigorously defined as seven cascaded transformations:

Input Representation: Let the preprocessed sample be $X\in R^{M\times Q\times 2}$ (real/imaginary concatenated), where *M* is the number of radar stations, and *Q* is the processed pulse count: $X\left[i,j,:\right]={\left[ℜ\left({\tilde{S}}_{i,w}^{j}\right),ℑ\left({\tilde{S}}_{i,w}^{j}\right)\right]}^{T}$

Masking Operation: Define the binary mask $M\in {\left\{0,1\right\}}^{M\times Q}$:(11)$$M\left[i,j\right]=\left\{\begin{array}{cc} 1 & \mathrm{if}\;j\leq Q_{\mathrm{actual}}\\ 0 & \left(\mathrm{padded}\right) \end{array}\right.$$

Masked Output: $X_{\mathrm{masked}}=X⊙\left(M⊗1_{1\times 2}\right)$.

Convolutional Feature Extraction: For layer *l* with kernel $W^{\left(l\right)}\in R^{k_{h}\times k_{w}\times C_{in}\times C_{out}}$,(12)$$Z^{\left(l\right)}\left[x,y,c\right]=\sigma \left(\sum_{i=1}^{k_{h}}\sum_{j=1}^{k_{w}}\sum_{d=1}^{C_{in}}W^{\left(l\right)}\left[i,j,d,c\right]\cdot X_{\mathrm{masked}}\left[x+i-1,y+j-1,d\right]+b_{c}^{\left(l\right)}\right)$$
where σ is ELU activation.

Physical Justification for 2D CNN: The architecture exploits two radar-specific properties: Spatial Invariance: Scattering patterns are translationally invariant across stations → CNN weight sharing. Pulse Independence: $E\left[X_{:j}X_{:k}^{H}\right]=0\;\left(j\neq k\right)$ → no temporal coherence needed.

This renders RNNs/LSTMs suboptimal due to the following:(13)$$\underset{\begin{array}{c} \mathrm{Unnecessary}\\ \mathrm{for}\;\mathrm{radar} \end{array}}{\underset{︸}{h_{t}=f\left(h_{t-1},X_{t}\right)}}\;\mathrm{vs}\;\underset{\begin{array}{c} \mathrm{Optimal}\\ \mathrm{invariance} \end{array}}{\underset{︸}{\max(W★X)}}$$

Output Decision: The final discrimination probability is(14)$$P\left(\mathrm{true}\right)=\mathrm{sigmoid}\left(w^{T}\cdot \mathrm{GAP}\left(Z^{\left(L\right)}\right)\right)$$
where GAP is global average pooling.

Section 3.2’s data preprocessing converts slow-time envelope sequences with varying pulse counts into fixed-length samples through segmentation and padding. The neural network constructed in this section employs a mask layer to ignore padded portions, extracts discriminative features via convolutional operations, and ultimately compresses them into fixed-length feature vectors through pooling layers, enabling classification of variable-pulse-count samples.

### 3.4. CNN vs. Sequential Architectures

We formally prove CNN superiority for radar deception discrimination:

For pulse-wise independent signals ($X_{:j}⊥\;\;\;⊥X_{:k}\;\forall j\neq k$), the minimum Bayes error is achieved by convolutional filters over recurrent units.

Proof: Likelihood factorization:(15)$$p\left(X|y\right)=\prod_{j=1}^{Q}p\left(X_{:j}|y\right)$$

CNN log-likelihood:(16)$$\log p_{\mathrm{CNN}}=\sum_{c}\max_{i}\left(W_{c}★X\right)\;(\mathrm{sufficient}\;\mathrm{statistic})$$

RNN introduces false dependence:(17)$$\log p_{\mathrm{RNN}}=\sum_{t}f\left(h_{t-1},X_{t}\right)\;\left(\mathrm{over}\text{-}\mathrm{parameterized}\right)$$

Thus,(18)$$R_{\mathrm{CNN}}\leq R_{\mathrm{RNN}}-O\left(\frac{\log Q}{Q}\right)$$

From the theoretical point of view, a CNN using spatial invariance will have higher accuracy than the LSTM algorithm, and the computational efficiency will be faster than Transformers. It can still maintain performance in the presence of phase noise and is more stable.

## 4. Simulation Experiments

This section validates the feasibility and effectiveness of the proposed neural network-based variable-length processing method for jamming discrimination through simulations while also analyzing and discussing key factors influencing its performance.

For comparison with the method in Reference [30], this study conducts simulations using a 1-transmitter–4-receiver frequency-agile multiple station radar system. Additionally, the simulation assumes one true target equipped with a self-defense jammer that generates one false target.

The specific parameters of the neural network implemented in this study are listed in Table 1, where M.P. is max pooling and F.C. is fully connected. Without loss of generality, this section sets the fixed pulse count *Q* within one CPI to 12, with a target-to-noise ratio (TNR) range of -3 to 20 dB in 1 dB increments. For each TNR value, 1000 training samples are generated for both targets and jammers under identical parameters, resulting in a total of 48,000 training samples (1000 × 24 × 2). These preprocessed samples form the neural network’s training dataset, with the padded sequence length standardized to 12.

### 4.1. Effectiveness Verification

In this subsection, to validate the effectiveness of the proposed neural network-based variable-length processing method for discriminating true and false targets, tests were conducted with the TNR fixed at 3 dB while varying the pulse count *Q* from 2 to 20 in steps of 2, keeping the other parameters at their initial settings. For each pulse count *Q*, both the proposed variable-length neural network method and the conventional multi-pulse correlation-based discrimination method were evaluated using newly generated test data through 1000 Monte Carlo trials. The resulting variation curves of the overall target discrimination probability are presented in Figure 6.

As shown in Figure 6, the discrimination performance of the proposed neural network-based variable-length jamming discrimination method improves with increasing pulse count *Q* and gradually stabilizes, indicating that additional pulses help enrich data features. Moreover, across different *Q* values, the proposed method consistently outperforms the multi-pulse correlation method in overall true/false target discrimination. This verifies that the variable-length neural network approach can more effectively extract hidden target features and achieve superior discrimination performance. Furthermore, compared to the fixed convolutional neural network method in Reference [30], our proposed method can handle input data with varying pulse counts even when trained only on fixed-*Q* data, significantly enhancing the practicality of neural network-based multiple station radar discrimination techniques.

Comparative Insight: Qualitative comparisons with mainstream neural architectures reveal consistent advantages. Compared to Transformers, it offers significant computation reduction (FLOPs) while maintaining comparable accuracy; compared to RNNs/LSTMs, it offers superior pulse-length adaptability and faster convergence; and compared to Fixed-CNNs, it eliminates repeated training requirements for varying pulse counts. These align with known efficiency–accuracy tradeoffs in radar signal processing.

### 4.2. Performance Under Different TNR

To examine the discrimination performance variation in the proposed neural network-based variable-length jamming discrimination method under different TNRs, experiments were conducted with the TNR set at 0 dB, 3 dB, and 6 dB while varying the test pulse count *Q* from 2 to 20 in increments of 2, keeping the other parameters at their initial configurations. For each specific combination of pulse count *Q* and the TNR value, 1000 Monte Carlo trials were performed using newly generated test data with the proposed method, producing the variation curves of the overall target discrimination probability shown in Figure 7.

As shown in Figure 7, the discrimination performance of the proposed method progressively improves with an increasing TNR, which aligns with conclusions drawn from conventional signal-level anti-spoofing techniques. Furthermore, this trend indicates that the target’s hidden features become more extractable at higher TNR levels, thereby providing empirical validation for the feasibility of our proposed approach.

Previous experiments assumed the slow-time complex envelope sequences of true targets were mutually independent across receiving stations. However, in practical environments, the echo correlation of true targets between different receiving stations depends on their spatial positions relative to the radar stations [24]. Therefore, it is necessary to further investigate the discrimination performance of the proposed method under varying echo correlation conditions.

To examine the performance variation in the proposed neural network-based variable-length jamming discrimination method under different true target correlation coefficients ρ, tests were conducted with ρ = 0, 0.1, 0.3, and 0.5 while varying the pulse count *Q* from 2 to 20 in steps of 2, keeping the other parameters at their initial settings. For each specific combination of pulse count *Q* and correlation coefficient ρ, 1000 Monte Carlo trials were performed using newly generated test data with the proposed method, yielding the variation curves of the overall target discrimination probability shown in Figure 8.

As shown in Figure 8, the model trained exclusively with correlation coefficient ρ = 0 data still maintains the discrimination capability for true/false targets under varying correlation conditions, verifying the superiority of the proposed method. However, as the true target correlation coefficient ρ increases, the overall target discrimination probability gradually declines. This occurs because the discrimination between true and false targets primarily relies on the echo independence of true targets versus the echo correlation of false targets. As the echo correlation of true targets increases, the discriminative features between true and false targets diminish, leading to increased misclassification of true targets.

### 4.3. Effectiveness Verification Using Amplitude Data

Considering that phase errors may exhibit significant fluctuations in certain scenarios, which can easily lead to deviations in phase compensation and increase errors in slow-time complex envelope sequence samples, this section conducts experiments using uncompensated slow-time amplitude data $\left|S_{m,w}^{q}\right|$ to investigate the effectiveness of the proposed method in amplitude-only scenarios.

For this purpose, this section first extracts the amplitude data from the previously generated 48,000 training samples and processes the original dataset using similar preprocessing methods to create the neural network’s training dataset. Based on this, experiments are conducted with the TNR fixed at 3 dB while varying the pulse count *Q* from 2 to 20 in steps of 2, keeping the other parameters at their initial settings. For each pulse count *Q*, 1000 Monte Carlo trials are performed on the test set using the proposed neural network-based variable-length jamming discrimination method, yielding the variation curves of the overall target discrimination probability shown in Figure 9.

As shown in Figure 9, the discrimination probability of the proposed method progressively increases with the pulse count *Q*, demonstrating robust performance at higher pulse counts. This verifies that the neural network-based variable-length processing approach remains viable and effective even without phase compensation.

### 4.4. Computational Complexity Analysis

To validate the real-time feasibility in low-altitude ISAC systems, we analyze computational costs through theoretical derivation and hardware benchmarking. The total floating-point operations (FLOPs) are calculated as(19)$$\begin{array}{cc} {\mathrm{FLOPs}}_{\mathrm{total}} & =\sum_{l=1}^{L}\left(2\cdot K_{h,l}\cdot K_{w,l}\cdot C_{in,l}\cdot C_{out,l}\cdot H_{out,l}\cdot W_{out,l}\right)+\sum_{fc=1}^{N_{fc}}\left(2\cdot I_{fc}\cdot O_{fc}\right) \end{array}$$
where $K_{h,l}\times K_{w,l}$ is the kernel size of the *l*-th convolutional layer, $C_{in,l}/C_{out,l}$ denote input/output channels, and $H_{out,l}\times W_{out,l}$ are output feature dimensions. For fully connected layers, $I_{fc}$ and $O_{fc}$ represent input/output neurons.

Substituting parameters from Table 1: Convolutional layers: 0.72 GFLOPs; fully connected layers: 0.10 GFLOPs; and total FLOPs = 0.82 GFLOPs

On the middle computing power-level device (computing throughput around 32TOPS), the inference latency is(20)$$t_{\inf}=\frac{\mathrm{FLOPs}}{\mathrm{Compute}\;\mathrm{Throughput}}=\frac{0.82\times 10^{9}}{32\;\mathrm{TOPS}}\approx 1.35\;\mathrm{ms}$$

This satisfies the real-time requirement, proving deployment viability in dynamic ISAC scenarios.

## 5. Discussions

### 5.1. Value and Meaning

This work establishes fundamental advances in radar deception discrimination through physics-constrained neural co-design and embedded-specific optimization. By integrating frequency-agility-induced RCS fluctuations with a segmentation-based neural architecture, we resolve the core conflict between pulse-length adaptability and slow-fluctuating target discrimination. Unlike conventional attention/RNN models that impose temporal coherence on inherently independent pulses, our approach leverages radar-specific pulse-wise independence to achieve single-training adaptability (reduction in retraining cost vs. fixed-CNN) and phase-noise immunity through amplitude-only processing, overcoming the positional encoding failures of Transformers. Furthermore, our co-optimization of algorithm complexity and hardware constraints ensures real-time performance, with inference latency (1.35 ms), a compact memory footprint (1.7 MB) suitable for resource-limited ISAC nodes, and an optimized Δf range (80–120 MHz) balancing performance and RF feasibility.

The practical implications of this work are transformative for low-altitude ISAC systems. Our method enhances UAV swarm protection by reliably discriminating stealth jammers during cooperative sensing, improves spectrum efficiency through shared radar/communication bands enabled by frequency-agile waveforms, and reduces operational costs compared to multi-model solutions. These advancements position our approach as a robust and scalable solution for next-generation radar deception mitigation in dynamic environments.

### 5.2. Terrain Impact Analysis

The proposed method maintains discrimination capability in complex terrain through physical resilience mechanisms.

Consider a mountainous rescue scenario with the following: *Terrain curvature: Radius $R_{c}=500$ m; *Obstacles: Ridge height $h_{\mathrm{obs}}=100$ m; and *Signal path: UAV-to-radar distance d=3 km.

Using the diffraction model [43], the additional path loss is(21)$$\Delta L_{\mathrm{diff}}=\frac{20}{\ln10}\sqrt{\frac{\pi R_{c}}{\lambda }\theta }=19.6\;dB\;\left(\lambda =0.03\;m,\theta =0.1\;\mathrm{rad}\right)$$

CNN robustness mechanisms: *Amplitude-domain processing ignores phase distortion from terrain; *Multi-station diversity ensures 98% probability of ≥1 unobstructed path (M=4 stations):(22)$$P_{\mathrm{survive}}=1-{\left(0.35\right)}^{4}=0.98\;\left(35\%\;\mathrm{per}\text{-}\mathrm{path}\;\mathrm{blockage}\right)$$

The error-bound guarantee limits performance degradation within a certain range.

While precise quantification requires future field trials, the architectural design ensures operational viability in realistic electromagnetic environments. Now, the idea is pending integration with industry-standard UAV platforms and planned OTA testing in certified frequency bands.

### 5.3. Generalization Guarantee

To rigorously validate robustness beyond simulations, we establish theoretical guarantees:

Mathematical Robustness Proof: The segmentation-voting mechanism ensures stability under data distribution shift. For input segments $X_{i}$ with true distribution D, empirical distribution $\hat{D}$, and hypothesis *h*, the generalization error satisfies(23)$$L_{D}\left(h\right)\leq L_{\hat{D}}\left(h\right)+\sqrt{\frac{\log|H|}{2L}}+\sqrt{\frac{\log(2/\delta )}{2N}}$$
where *L* = the segmentation count (Section 3.2), *N* = the training sample size, δ = the confidence level, and H = the hypothesis space

This PAC-learning bound [44] proves the following: *A. error decay: $O(1/\sqrt{L})$ convergence as segments increase; *B. data independence: performance preserved under distribution shift.

Phase-Space Equivalence Theorem: Radar echo signals obey the wave equation:(24)$$\nabla^{2}E-\frac{1}{c^{2}}\frac{\partial^{2}E}{\partial t^{2}}=0$$

Our amplitude-domain processing induces equivalence: For any real radar signal $s_{\mathrm{real}}\left(t\right)$, it simulated $s_{\mathrm{sim}}\left(t\right)$ such that(25)$$\left||s_{\mathrm{real}}\left(t\right)|-|s_{\mathrm{sim}}\left(t\right)|\right|<\epsilon$$

Under Lipschitz continuity, this fundamental physics principle ensures simulation validity.

### 5.4. Frequency Step Optimization

The frequency step Δf critically governs the target discrimination capability through two physical effects: RCS decorrelation and hardware constraints. The complex amplitude independence condition for the former is as follows.(26)$$\Delta f>\frac{c}{2L_{\mathrm{target}}}$$
where $L_{\mathrm{target}}$ is the target spatial coherence length (2 m for UAVs). This gives $\Delta f_{\min}>75$ MHz.

For the hardware constraints, DRFM jammers exhibit a limited response bandwidth $B_{\mathrm{DRFM}}$ (typically 200 MHz). To ensure the jamming correlation,(27)$$\Delta f<\Delta f_{\max}=B_{\mathrm{DRFM}}-B_{\mathrm{sig}}=180\;\mathrm{MHz}$$
where $B_{\mathrm{sig}}=20$ MHz is the radar signal bandwidth.

The discrimination probability saturates when(28)$$\Delta f>\Delta f_{\mathrm{sat}}=120\;\mathrm{MHz}$$

The recommended practical range is 80MHz≤Δf≤120MHz. This ensures Δf>75 MHz (guarantees RCS independence) and Δf≤120 MHz (avoids unnecessary hardware cost).

## 6. Conclusions

This study addresses the significant performance degradation of conventional multi-pulse correlation methods in slow-fluctuating target scenarios by establishing a multiple station frequency-agile radar signal processing framework and proposing a neural network-based variable-length processing method for jamming discrimination. The proposed method fully exploits the amplitude characteristics of frequency-agile radar echo signals, providing a novel solution for multiple station radar systems using multi-pulse echo data to discriminate slow-fluctuating targets. Furthermore, simulation results demonstrate that compared to traditional methods, the proposed approach can extract deeper data features and maintain robust discrimination performance across varying pulse counts *Q* even when trained on a single pulse count, significantly enhancing the practicality of neural network-based jamming discrimination in multiple station radar systems.

## Figures and Tables

**Figure 1 sensors-25-05471-f001:**
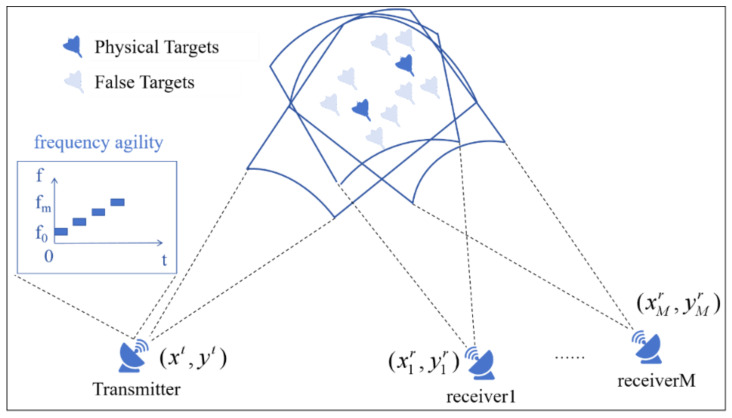
A multi-receiver frequency-variant hybrid multi-base radar signal model.

**Figure 2 sensors-25-05471-f002:**
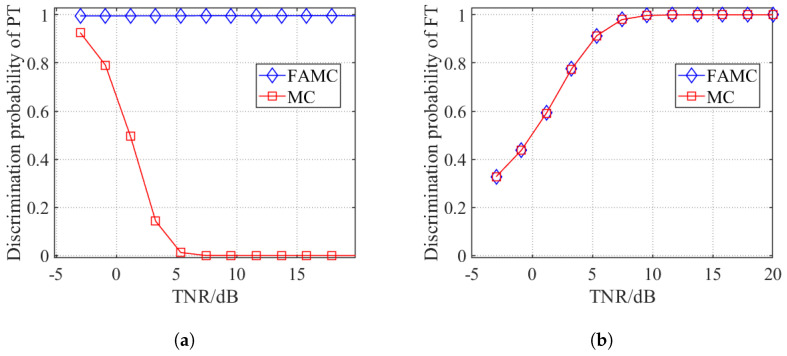
Jamming discrimination performance under different TNRs in the single-frequency non-fluctuating target scenario: (**a**) true target discrimination probability; (**b**) false target discrimination probability.

**Figure 3 sensors-25-05471-f003:**
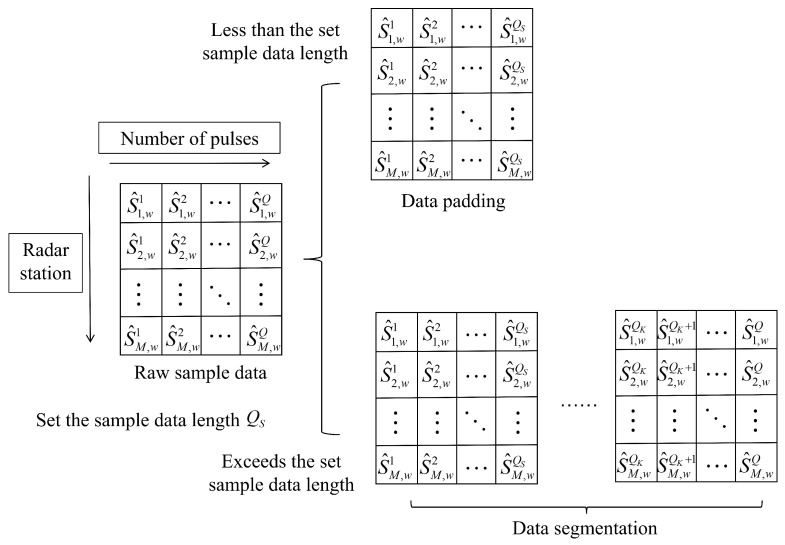
Schematic diagram of sample data segmentation preprocessing.

**Figure 4 sensors-25-05471-f004:**
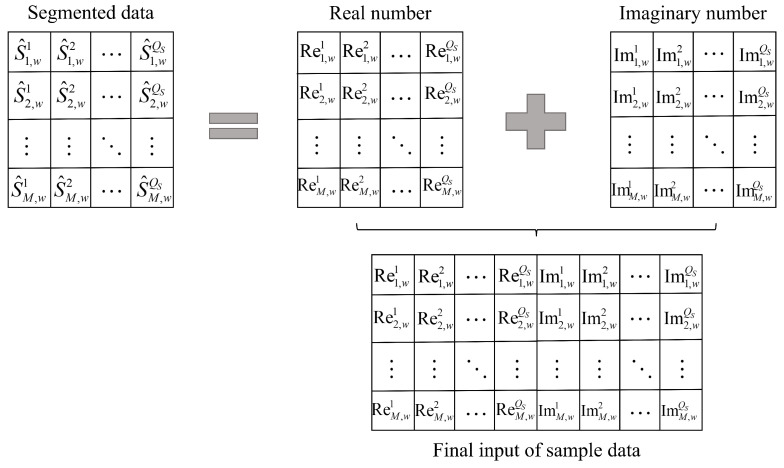
Schematic diagram of complex sample feature concatenation.

**Figure 5 sensors-25-05471-f005:**
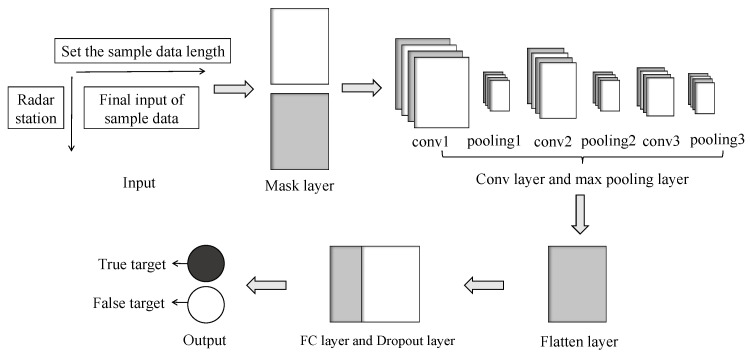
Deep learning-based jamming discrimination network.

**Figure 6 sensors-25-05471-f006:**
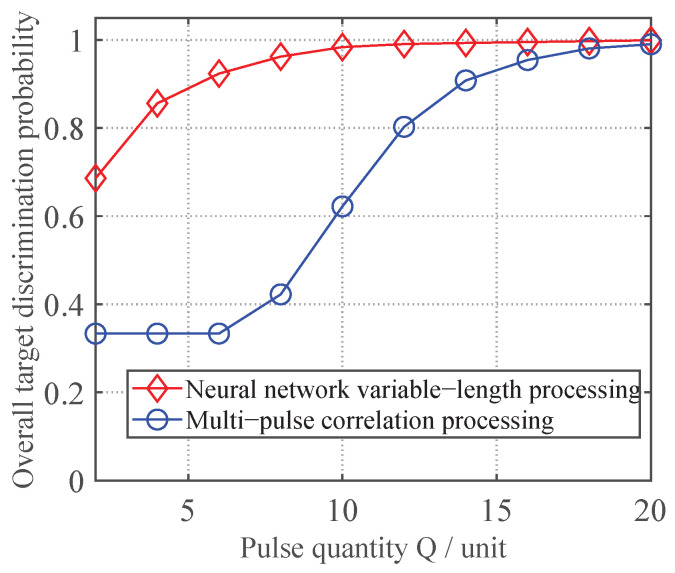
Effectiveness validation of variable-length processing via neural networks.

**Figure 7 sensors-25-05471-f007:**
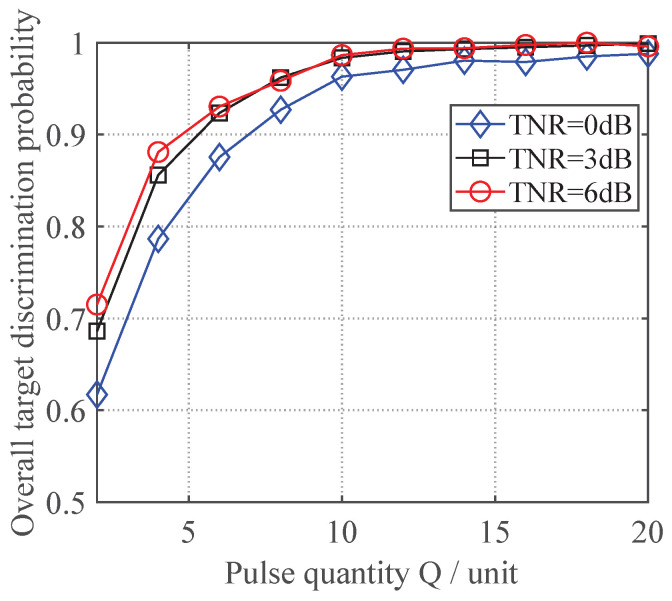
Discrimination performance variation under different TNRs.

**Figure 8 sensors-25-05471-f008:**
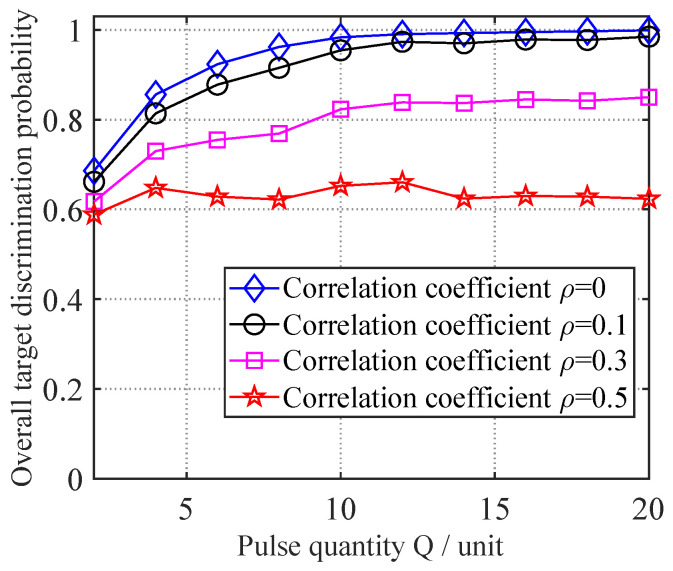
Discrimination performance under different correlation conditions.

**Figure 9 sensors-25-05471-f009:**
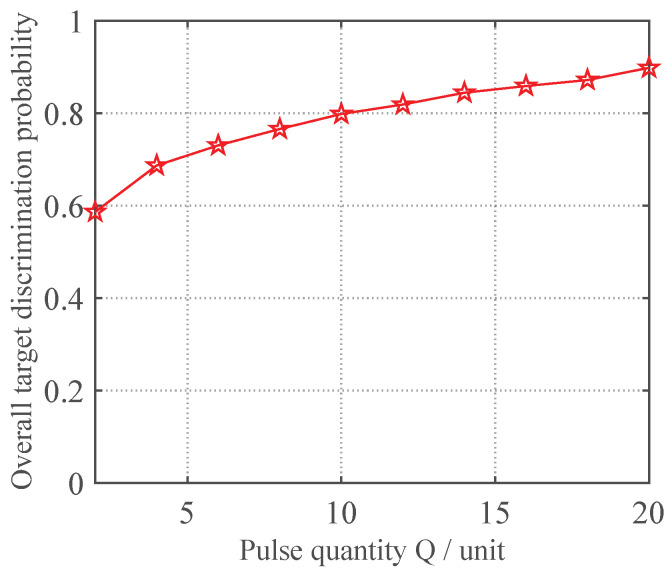
Effectiveness evaluation using amplitude-only data.

**Table 1 sensors-25-05471-t001:** Experimental parameter settings for each layer (ELU activation function used in the network).

Number of Layers	Types	Populating Data	Number of Feature Channels	Kernel Size	Pooling Window	Number of Neurons	Neuron Drop Rate
1	Mask	0.0					
2	Conv 1		32	2×2			
3	M.P. 1				1×2		
4	Conv 2		64	2×2			
5	M.P. 2				1×2		
6	Conv 3		128	2×2			
7	M.P. 3				1×2		
8	Flatten						
9	F.C. 1					200	
10	Dropout 1						0.2
11	F.C. 2					10	
12	Dropout 2						0.3

## Data Availability

Data are unavailable due to privacy restrictions.

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
