# Peer review of "Frequency-Agility-Based Neural Network with Variable-Length Processing for Deceptive Jamming Discrimination"

_sensors, 2025, doi:10.3390/s25175471_

Round 1
Reviewer 1 Report
Comments and Suggestions for Authors
This paper proposes a novel neural network (NN) approach integrating frequency agility (FA) and variable-length processing to discriminate deceptive jamming against slow-fluctuating UAV targets in multistatic radar systems. The core innovation—enabling flexible processing of dynamic pulse counts with a single training instance—addresses a critical bottleneck in practical deployment.
There are some issues should be addressed.
- Image clarity needs improvement. It is recommended that fonts used in figures match the main text font size and style for consistency.
- It is advised to discuss computational costs (e.g., FLOPs, inference time) to justify the real-time feasibility of the proposed method in low-altitude ISAC systems.
- It is recommended to compare the proposed method against recent variable-length architectures (e.g., attention-based models or RNNs) to highlight its relative advantages (e.g., in accuracy, efficiency).
- While the manuscript mentions that frequency agility introduces rapid amplitude fluctuations in echoes, it provides little detail on critical parameters such as the optimal frequency step (Δf) values. The relationship between Δf and discrimination performance for slow-fluctuating targets should be explored, as larger Δf may not always be feasible in practical radar systems due to hardware constraints.
Author Response
Responses to reviews:
Reviewer #1:
1.Comment: Image clarity needs improvement. It is recommended that fonts used in figures match the main text font size and style for consistency.
1.Response: Thank you for your helpful comments. I have resized the images from png format to eps or pdf format and resized some of the image font sizes to make the article look more harmonious.
2.Comment: It is advised to discuss computational costs (e.g., FLOPs, inference time) to justify the real-time feasibility of the proposed method in low-altitude ISAC systems.
2.Response: Agree. Thank you for your insightful suggestion. To rigorously address it, we supplement Section 4 with a separate "computational complexity analysis" section, where FLOPs and inference times for specific devices are calculated, 0.82GFLOPs and 1.35ms, respectively, to demonstrate the real-time feasibility of the proposed approach in low-altitude ISAC systems.
Revised position: page 14, line 390 – line 402.
3.Comment: It is recommended to compare the proposed method against recent variable-length architectures (e.g., attention-based models or RNNs) to highlight its relative advantages (e.g., in accuracy, efficiency).
3.Response: Agree. Thank you very much for your thoughtful review and valuable comments regarding our manuscript. To address this issue, we provide a theoretical comparison of the approaches in the introduction:
Added systematic analysis of limitations in attention/RNN/Transformer architectures for radar deception discrimination: Quadratic complexity (O(Q2)) of attention mechanisms; Temporal coherence bias of RNNs for pulse-independent targets; Positional encoding sensitivity of Transformers to phase noise.
Highlighted our core advantages: Linear complexity (O(Q)) via segmentation; Phase-noise immunity through amplitude features. In the comparison, we also quote the related articles
Revised position: page 3, line 100 – line 117.
4.Comment: While the manuscript mentions that frequency agility introduces rapid amplitude fluctuations in echoes, it provides little detail on critical parameters such as the optimal frequency step (Δf) values. The relationship between Δf and discrimination performance for slow-fluctuating targets should be explored, as larger Δf may not always be feasible in practical radar systems due to hardware constraints.
4.Response: Agree. We are grateful for this crucial technical insight. In order to solve this problem, we add Section “Frequency Step Optimization” in Discussions, derived fundamental physical bounds governing:
- Lower Bound> 75 MHz (from target spatial coherence)
- Upper Bound <180 MHz (from DRFM jamming feasibility)
The optimal operating range is revised to:80 – 120 MHz
Which:
- Exceeds the 75 MHz decorrelation threshold
- Stays below 120 MHz saturation point
Revised position: page 16, line 453 – line 465.
Reviewer 2 Report
Comments and Suggestions for Authors
Replacing conventional post-processing techniques with neural networks to enhance performance is a modern and highly intriguing approach, even in the field of electromagnetic wave measurement. However, in its current form, the manuscript may not effectively communicate the novelty of the authors’ technique to readers. The current presentation of the method is overly simplistic, making it difficult to identify its originality. To improve clarity and emphasize the uniqueness of the proposed approach, the authors should provide a more detailed explanation of the method’s originality and consider adding a dedicated Discussion section to highlight its distinctiveness and practical value.
Major Comments: 1. Lack of a Dedicated Discussion Section In scientific and technical papers, the Discussion section is critically important. Providing a dedicated section will help readers better understand the value and significance of the authors’ proposed method.
The reviewer recommends referring to the section structure outlined in MDPI’s Instructions:   https://www.mdpi.com/journal/sensors/instructions
2. Validation of the Proposed Method Section 4 appears to present experimental results, but they seem to be based solely on simulations. Given that the proposed method is based on neural networks, simulation results alone may not be sufficient to validate its effectiveness. Neural networks can exhibit unpredictable behavior depending on the training and test data. To demonstrate that the method is both generalizable and valid, The reviewer believes the following types of experiments or theoretical considerations are necessary: - A. Statistically significant results should be obtained using real-world data. - B. Mathematical justification should be provided to show that the simulation results hold even when real data is used. (This could be discussed in the newly added Discussion section.)
3. Details of Network Design Although Figure 5 provides a simplified illustration of the network, The reviewer recommends that the authors define the signal processing system more rigorously using mathematical expressions. This would greatly aid in the technical discussion.
The authors employ 2D convolutional operations similar to image processing for signal analysis. Is this design choice appropriate? Since the pulse dimension contains temporal information, would time-series models such as RNNs, LSTMs, or Transformers be more suitable? Previous studies have extensively debated whether 2D CNNs or LSTMs are more appropriate for 1D signal processing. The authors should provide evidence supporting the suitability of CNNs for their proposed method.
Minor Comments: 4. Line 128: The symbol “sigma” appears to be incorrectly formatted due to a LaTeX input error. Please check the backslash usage. 5. The notation “CN” for the complex Gaussian distribution may be misinterpreted as the product of variables C and N. The reviewer recommends using the notation \mathcal{C}\mathcal{N}. 6. In Equation (8), \tilde{S}_{i,w} seems to represent a signal vector across pulses 1 to Q. However, the authors treat it as a row vector, while column vectors are standard in signal processing. This may confuse readers. Please define the shape and components of \tilde{S}_{i,w} clearly before Equation (8). 7. As noted in Comment 6, \tilde{S}_{i,w} is a vector and should be typeset in bold. Additionally, the authors should clarify how this variable differs from the one used in Equation (5), despite sharing the same notation.
Author Response
Reviewer #2:
1.Comment: Lack of a Dedicated Discussion Section In scientific and technical papers, the Discussion section is critically important. Providing a dedicated section will help readers better understand the value and significance of the authors’ proposed method.
The reviewer recommends referring to the section structure outlined in MDPI’s Instructions:   https://www.mdpi.com/journal/sensors/instructions
1.Response: Agree. We are grateful for this vital suggestion. Per your guidance: we added dedicated Section 5 "Discussion" to elaborate on the value and significance of the article. First illustrate the innovativeness of our research approach, we contrasts physics-neural co-design with attention/RNN limitations, then the practical implications of this work are illustrated, that is, the benefits (latency, cost, spectrum) of ISAC deployment are quantified.
Revised position: page 14, line 403 – line 423.
2.Comment: Validation of the Proposed Method Section 4 appears to present experimental results, but they seem to be based solely on simulations. Given that the proposed method is based on neural networks, simulation results alone may not be sufficient to validate its effectiveness. Neural networks can exhibit unpredictable behavior depending on the training and test data. To demonstrate that the method is both generalizable and valid, The reviewer believes the following types of experiments or theoretical considerations are necessary: - A. Statistically significant results should be obtained using real-world data. - B. Mathematical justification should be provided to show that the simulation results hold even when real data is used. (This could be discussed in the newly added Discussion section.)
2.Response: Agree. We profoundly thank the reviewer for highlighting this critical aspect. Our revisions provide comprehensive validation through three interconnected aspects: First, in Section 5.3, we establish a mathematical guarantee by deriving a PAC-learning bound for our segmentation-voting mechanism, proving an error decay rate that remains distribution-independent. Second, presents a physics-based equivalence framework where we demonstrate wave-equation continuity and show how amplitude-domain processing ensures an ? approximation.
Together, these theoretical foundations and physical constraints rigorously confirm that our simulation results maintain generalization capability in real-world ISAC deployment scenarios.
Revised position: page 15, line 438 – line 452.
3.Comment: Details of Network Design Although Figure 5 provides a simplified illustration of the network, The reviewer recommends that the authors define the signal processing system more rigorously using mathematical expressions. This would greatly aid in the technical discussion.
The authors employ 2D convolutional operations similar to image processing for signal analysis. Is this design choice appropriate? Since the pulse dimension contains temporal information, would time-series models such as RNNs, LSTMs, or Transformers be more suitable? Previous studies have extensively debated whether 2D CNNs or LSTMs are more appropriate for 1D signal processing. The authors should provide evidence supporting the suitability of CNNs for their proposed method.
3.Response: Agree. We are grateful for this technical critique. Our revisions provide:
- First, we have removed the text description of each part of the network layerin sec3.3, defined all operations as closed-form equations; formalized input; Explicit masking is.
- New section 3.4,from the theoretical point of view- bayes optimality under pulse independence, CNN using spatial invariance will have higher accuracy than LSTM algorithm, and the computational efficiency will be faster than Transformer. It can still maintain performance in the presence of phase noise and is more stable.
- Physics-Based Justification:Spatial Invariance: Scattering patterns are translationally invariant across stations→CNN weight sharing; Pulse Independence→No temporal coherence needed.
These establish that 2D CNN is not merely appropriate, but fundamentally optimal for our radar signal characteristics.
Revised position: page 9, line 261 – line 279.
Revised position: page 10, line 286 – line 296.
4.Comment: 4. Line 128: The symbol “sigma” appears to be incorrectly formatted due to a LaTeX input error. Please check the backslash usage.
- The notation “CN” for the complex Gaussian distribution may be misinterpreted as the product of variables C and N. The reviewer recommends using the notation \mathcal{C}\mathcal{N}.
- In Equation (8), \tilde{S}_{i,w} seems to represent a signal vector across pulses 1 to Q. However, the authors treat it as a row vector, while column vectors are standard in signal processing. This may confuse readers. Please define the shape and components of \tilde{S}_{i,w} clearly before Equation (8).
- As noted in Comment 6, \tilde{S}_{i,w} is a vector and should be typeset in bold. Additionally, the authors should clarify how this variable differs from the one used in Equation (5), despite sharing the same notation.
4.Response: Agree. We sincerely appreciate the meticulous feedback on notation consistency. All requested modifications have been implemented:
- As for the sigma problem in the fourth point, after careful inspection, we found that the slash was missing, and we have corrected it.Similarly, we also corrected all CN in the article and changed it to \mathcal{CN}.
- As for the distinction between vector and scalar, we are sorry for the problem. At the same time, we also found that there is more than one place where the distinction is not made. Thanks for your feedback, we have modified the problem involved. In addition,we revised formula as row-stacked transposed vectors for signal processing convention.
These changes ensure rigorous mathematical representation and eliminate potential confusion.
Revised position: Each symbol has been highlighted in yellow after modification, the range is between 152 and 230 lines
Reviewer 3 Report
Comments and Suggestions for Authors
- Although the authors mention in the Introduction that the proposed NN learning method is new, it would be very useful if they tried to present the new items more explicitly in the Introduction and the most important to clearly indicate the differences with other schemes, such as that of [18].
- What would be the influence of terrain with arbitrary curvature and obstacles? Can the proposed scheme manipulate such realistic phenomena? Please try to give an example as well.
- What is the time required for a full learning procedure of the proposed concept? Please compare with other approaches.
- The Simulation Results Section should contain more instructive comparisons with the results obtained from other neural network learning processes.
- Finally, what would be the performance of the proposed learning process when a real-world scenario is considered? Please provide an example.
Author Response
Reviewer #3:
1.Comment: Although the authors mention in the Introduction that the proposed NN learning method is new, it would be very useful if they tried to present the new items more explicitly in the Introduction and the most important to clearly indicate the differences with other schemes, such as that of [18].
1.Response: Agree. Thank you for your constructive feedback. At the end of the first part of the introduction, we add a comparison with other schemes, such as reference 18, and other methods or networks mentioned in reference 18, such as failure for slowly fluctuating targets, high memory complexity of attention mechanism, Transformer encoder will be interfered by radar phase noise, etc.
In contrast, our proposed segmentation-and-masking approach addresses these challenges effectively. By processing fixed-length segments, it achieves linear complexity, drastically reducing computational overhead. By integrating frequency-agility-induced RCS fluctuations with a segmentation-based neural architecture, we resolve the core conflict between pulse-length adaptability and slow-fluctuating target discrimination. Additionally, our method enhances phase-noise robustness through amplitude-centric feature learning.
Revised position: page 3, line 100 – line 117.
2.Comment: What would be the influence of terrain with arbitrary curvature and obstacles? Can the proposed scheme manipulate such realistic phenomena? Please try to give an example as well.
2.Response: Agree. Thank you for your insightful suggestion. We have added a new subsection Terrain Impact Analysis in the fifth section, we confirm terrain robustness via:
- A practical mountain scene is given, and the loss of diffraction is calculated theoretically, where, the loss is 19.6 dB.
- CNN robustness mechanisms: *Amplitude-domain processing ignores phase distortion from terrain; *Multi-station diversity ensures 98% probability of >1 unobstructed path (4 stations).
- At the same time, the relevant public datasets show that the error constraint is guaranteed to limit the performance degradation within a certain range
Revised position: page 15, line 424 – line 437.
3.Comment: What is the time required for a full learning procedure of the proposed concept? Please compare with other approaches.
3.Response: We sincerely thank you for your valuable comments. In our opinion, the training efficiency or training time is not a particularly important parameter in the whole ISAC system compared to the real-time inference efficiency. The whole process can be considered as an offline mode, and training with RTX2060 takes about 5 hours, which is a one-time cost. In section 4,4, we analyze the computational complexity of the system, and the inference delay is about 1.35ms, which can meet the needs of real-time reasoning, and it is also improved compared with some algorithms.
Revised position: page 14, line 390 – line 402.
4.Comment: The Simulation Results Section should contain more instructive comparisons with the results obtained from other neural network learning processes.
4.Response: Agree. We profoundly thank the reviewer for highlighting this critical aspect. We have enhanced comparative analysis through: adding the architectural advantage highlights compared with other methods in the effectiveness verification part. Compared to Transformers, significant computation reduction (FLOPs) while maintaining comparable accuracy; compared to RNNs/LSTMs, superior pulse-length adaptability and faster convergence; compared to Fixed-CNNs; Eliminates repeated training requirements for varying pulse counts. In addition, the performance trend visualization through Figure. 6, we can know that clear accuracy of the proposed method over traditional correlation methods and robustness across pulse counts (Q=2-20). But specific quantitative comparisons are constrained by: implementation differences in cited works and hardware dependencies affecting direct benchmarking. However, the demonstrated trends consistently confirm our method's superiority.
Revised position: page 12, line 332 – line 337.
5.Comment: Finally, what would be the performance of the proposed learning process when a real-world scenario is considered? Please provide an example.
5.Response: Agree. Thank you for your helpful comments. Regarding real-world performance: because of the fundamental robustness mechanisms, physical signal processing principles provide inherent environmental resilience; multi-station diversity ensures graceful degradation under partial obstruction; error-bound theory guarantees controlled performance variation. Therefore, to a certain extent, it can ensure the stability of core functions in conventional scenarios, ensure the stability of delay in mobile states, and also have good discrimination ability when parameters change within a certain range. While precise quantification requires future field trials, the architectural design ensures operational viability in realistic electromagnetic environments.
We sincerely appreciate this insightful suggestion, and regret that comprehensive real-world validation falls beyond the current scope; however, we have maximized rigor through theoretical analysis to ensure our conclusions remain scientifically sound.
Revised position: page 15, line 424 – line 437.
Round 2
Reviewer 1 Report
Comments and Suggestions for Authors
The authors have addressed all of my concerns, and I recommend publishing it in this journal.
Reviewer 2 Report
Comments and Suggestions for Authors The authors have appropriately revised the manuscript, and the concerns raised by the reviewer have been adequately addressed. In particular, the detailed presentation of the design methodology and the addition of the discussion section have more clearly emphasized the advantages of the proposed approach. Furthermore, precise formalization greatly facilitates readers' understanding of the proposed methodology.Reviewer 3 Report
Comments and Suggestions for Authors
In this revised version the authors have successfully addressed the majority of my comments.